# Cement-Based Grouting Materials Modified with GO/NS Hybrids

**DOI:** 10.3390/ma18214820

**Published:** 2025-10-22

**Authors:** Longfei Lu, Guoxiang Yang, Yan Ai, Jingkai Qu, Jinrui Duan, Kun Yang, Wenbin Sun

**Affiliations:** School of Engineering and Technology, China University of Geosciences (Beijing), Beijing 100083, China; 2002230058@email.cugb.edu.cn (L.L.); 3002230012@email.cugb.edu.cn (Y.A.); 3002230036@email.cugb.edu.cn (J.Q.); 3002240053@email.cugb.edu.cn (J.D.); 2002230060@email.cugb.edu.cn (K.Y.); 2002230057@email.cugb.edu.cn (W.S.)

**Keywords:** graphene oxide, nano-silica, cement-based grout, mechanical properties, synergistic effect

## Abstract

This study systematically investigates the effects of individual and combined incorporation of graphene oxide (GO) and nano-silica sol (NS) on the macroscopic properties and microstructure of cement-based grouting materials, with emphasis on their synergistic mechanisms. A series of macroscopic tests including setting time, fluidity, bleeding rate, and mechanical strength were conducted, complemented by multi-scale microstructural characterization techniques such as scanning electron microscopy (SEM), X-ray diffraction (XRD), mercury intrusion porosimetry (MIP), and Fourier-transform infrared spectroscopy (FTIR). The results demonstrate that both NS and GO effectively reduce setting time and bleeding rate while enhancing mechanical strength; however, NS exhibits a more pronounced adverse effect on fluidity compared to GO. The hybrid system displays a distinct transition from synergy to antagonism: under low-dosage co-incorporation (2 wt% NS + 0.01 wt% GO), the flexural and compressive strengths increased by 13.5% and 45.5%, respectively, relative to the reference group. Microscopic analysis revealed that the synergistic interaction between the pozzolanic effect of NS and the templating effect of GO under this condition optimizes hydrate morphology and pore structure, leading to enhanced performance. Conversely, excessive dosage of either component induces agglomeration, resulting in microstructural deterioration and performance degradation. This study establishes optimal dosage ranges and combination principles for NS and GO in cement-based materials, providing a theoretical foundation for designing high-workability and high-strength cementitious composites.

## 1. Introduction

Cement-based composites are fundamental materials widely used in modern construction and major engineering projects, including civil infrastructure, hydraulic facilities, and transportation systems [1]. However, their inherent brittleness, low tensile strength, and susceptibility to cracking significantly limit their application potential and service life, particularly in extreme environments and high-performance structures. From a micromechanical perspective, the mechanical properties and durability of cementitious materials are largely governed by the composition and morphological distribution of hydration products, as well as the characteristics of the pore structure [2]. Therefore, controlling the hydration process to optimize microstructure has become a crucial strategy for developing high-performance cement-based materials.

The rapid development of nanotechnology and materials science has introduced the use of nanomaterials as modifiers in cementitious matrices, representing a major research frontier [3,4,5,6]. Among these, graphene oxide (GO) and nano-silica (NS) have attracted significant attention due to their unique physicochemical properties and substantial potential for material enhancement [7,8,9]. GO possesses a distinctive two-dimensional layered structure and exceptional theoretical mechanical strength. Its surface is rich in oxygen-containing functional groups (e.g., hydroxyl, epoxy, and carboxyl groups), which can effectively adsorb cement hydration ions and serve as nucleation sites for hydration products, thereby accelerating hydration [10,11,12]. Research by Lv et al. [13,14] also indicated that GO can inhibit the propagation of microcracks through a bridging effect, markedly enhancing the material’s strength and toughness. Nonetheless, the practical application of GO faces considerable challenges: Zhang et al. [15] observed that its high specific surface area and strong π–π conjugation effects cause GO sheets to agglomerate readily, making uniform dispersion within the cement matrix difficult. This not only reduces its reinforcing efficiency but may also introduce microdefects. Moreover, experimental studies by Li et al. [16] confirmed that incorporating GO negatively affects the rheological properties of fresh paste, resulting in a significant loss of fluidity, which is detrimental to practical workability.

On the other hand, NS consists of colloidal dispersions of nano-sized amorphous silica (SiO_2_) particles. Vijayan et al. [17] provided a systematic review of the pozzolanic effect of NS, highlighting its ability to react with calcium hydroxide (CH), a primary hydration product, to form additional calcium silicate hydrate (C–S–H) gel. This process not only refines the microstructure of the interfacial transition zone (ITZ) but also effectively fills nano-scale pores, significantly improving the compactness and durability of the system. However, the use of NS also has limitations: Qing et al. [18] found that the high specific surface area of nano-SiO_2_ particles greatly increases water demand, creating a strong dependence on high-range water reducers to maintain adequate workability. Additionally, Ji [19] noted that the pozzolanic reaction rate of NS is relatively slow, resulting in limited early-age strength contribution. Excessive incorporation may lead to particle agglomeration, and unreacted particles may remain as inert fillers, potentially impairing the microstructure.

Notably, recent studies suggest that GO and NS may exhibit theoretical potential for synergistic modification [20,21,22,23]. Liu et al. [23] proposed that GO could provide a template for the ordered growth of C–S–H gel, while the gel products resulting from the NS reaction could encapsulate and anchor GO sheets. This synergistic interaction promotes the formation of a denser and tougher three-dimensional network composite structure, potentially overcoming the performance limitations of single-modifier systems. This concept is supported by findings in other composite material systems; for example, Bouibed et al. [22] demonstrated in studies on epoxy resins and coating sealants that GO/SiO_2_ hybrids produce a “1 + 1 > 2” synergistic effect, substantially enhancing mechanical strength and corrosion resistance.

Despite considerable progress in single nanomaterial modification, research on GO/NS hybrid systems remains limited [24,25,26,27]. Mohammed et al. [26] emphasized that most existing studies focus on the influence of individual materials on final mechanical properties, while systematic investigations into the synergistic variations in key fresh-state properties (e.g., fluidity, stability, and setting time) within composite systems are still lacking. More importantly, from a micromechanical perspective, Li et al. [27] pointed out the scarcity of systematic research on how GO and NS collectively affect the hydration process, product morphology, pore structure evolution, and interface properties. There is a critical need for in-depth studies that reveal the multi-scale “physicochemical” synergistic mechanisms.

Against this background, the present study systematically investigates the individual and synergistic effects of GO and NS on the macroscopic properties and microstructure of cement-based composites. Using PO 42.5 ordinary Portland cement as the base material, experiments were conducted to evaluate the fluidity, bleeding rate, setting time of fresh paste, and the compressive/flexural strength of hardened paste, with varying dosages of GO and NS. Scanning electron microscopy (SEM), X-ray diffraction (XRD), mercury intrusion porosimetry (MIP), and Fourier-transform infrared spectroscopy (FTIR) were employed to analyze the interaction mechanisms and modification effects of GO and NS across four aspects: micromorphology, phase composition, pore structure, and chemical bonding. This research aims to address three fundamental questions: (1) the dose–response relationships for NS and GO individually; (2) the synergy-to-antagonism transition threshold in hybrid systems; and (3) the structure-property relationships between multi-scale structural evolution and macroscopic performance. The findings are expected to provide essential data support for the compound application of nanomaterials in cementitious matrices and offer valuable guidance for designing cement-based materials with high workability, strength, and toughness.

## 2. Materials and Methods

### 2.1. Raw Materials

The materials used in this study included Portland cement (P.O 42.5), colloidal nano-silica (NS), an aqueous dispersion of graphene oxide (GO), and a polycarboxylate ether (PCE)-based superplasticizer. All materials were thoroughly characterized to ensure the accuracy and reproducibility of the experimental results.

1.Cement:

Ordinary Portland cement (P·O 42.5 grade) was obtained from Anhui Yicheng Conch Cement Co., Ltd. (Wuhu, China). Its chemical composition, as determined by X-ray fluorescence (XRF) spectroscopy, is presented in Table 1. The fineness of the cement, expressed as the residue on a 45 μm sieve, was measured to be 10.9%.

2.Nano-silica (NS):

An alkaline colloidal nano-silica solution was supplied by Qingdao Ronghaipolyde Fine Chemical Co., Ltd. (Qingdao, China). The key physical and chemical properties of the NS are summarized in Table 2.

3.Graphene oxide (GO):

An aqueous graphene oxide dispersion, synthesized via a modified Hummers’ method, was provided by Suzhou Tanfeng Technology Co., Ltd. (Suzhou, China). The dispersion had a concentration of 10 mg/mL and a purity exceeding 90%, exhibiting a brownish-yellow to black appearance. The detailed characteristics of the GO nanosheets are listed in Table 3.

4.Superplasticizer:

A commercial polycarboxylate ether (PCE)-based superplasticizer with a solid content of 20% was used to ensure sufficient workability of the fresh mixtures.

### 2.2. Sample Preparation

A constant water-to-cement (w/c) ratio of 0.6 was used as the baseline to systematically evaluate the individual and synergistic effects of nano-silica (NS) and graphene oxide (GO) on the properties of cement-based grouting materials. A total of 14 mix designs were formulated, as specified in Table 4. The mixtures were divided into four series:

Series A (Reference): Control group, containing only 0.6 wt% superplasticizer.

Series B (NS-only): Mixes with NS content ranging from 2 wt% to 5 wt%.

Series C (GO-only): Mixes with GO content ranging from 0.01 wt% to 0.04 wt%.

Series D (NS/GO Hybrid): Mixes with combined NS and GO, designed to examine their interaction and potential synergistic effects at various ratios.

The dosages of all additives were calculated based on the mass of cement.

All mixing and sample preparation procedures were performed in a controlled laboratory environment maintained at 25 ± 2 °C. Raw materials were accurately weighed using an electronic balance. A strict mixing sequence was followed: deionized water, the superplasticizer (PCE), the GO dispersion, and the NS colloid were successively introduced into a high-speed slurry mixer (model: SYJ-10, Shandong Luda Testing Instrument Co., Ltd., Taian, China). The mixture was initially homogenized at 400 rpm for 2 min. Cement was then gradually added, and mixing continued at the same speed for an additional 5 min to obtain a homogeneous and stable fresh paste.

The resulting grout was cast into pre-assembled molds (40 mm × 40 mm × 160 mm). To minimize air entrapment, the molds were lightly vibrated on a vibrating table each time they were filled to approximately one-third of their height. After casting, the molds were covered and transferred to a standard curing room (20 ± 1 °C, relative humidity ≥ 90%) for 24 h of initial setting. After demolding, the specimens were continuously cured in lime-saturated water until the designated testing ages (3, 7, and 28 days) for subsequent mechanical property tests and microstructural analysis. A flowchart illustrating the sample preparation process is shown in Figure 1.

### 2.3. Testing Methods

1.Setting Time

The initial and final setting times of the fresh paste were measured using a Vicat apparatus according to the standard test method [3]. The sample was stored in a constant-temperature chamber (20 ± 1 °C, 90% RH) starting from the time of water addition. The initial setting time was recorded when the penetration depth of the needle reached 4 ± 1 mm above the base plate. The final setting time was defined as the time at which the needle no longer left a visible circular impression on the paste surface.

2.Fluidity (Flowability)

The fluidity of the fresh mixture was evaluated by measuring its flow diameter using the diffusion disc method [28]. Specifically, the freshly mixed paste was poured into a truncated conical mold placed centrally on a flat glass plate. The mold was then lifted vertically, allowing the paste to spread freely. After the flow stabilized, the final diameter was measured along two perpendicular directions. The average of these two values was reported as the flow value, with an accuracy of 1 mm.

3.Bleeding Rate

The stability of the fresh mixture was assessed based on its bleeding rate. A 200 mL graduated cylinder was filled with the freshly prepared grout, sealed, and left undisturbed at room temperature. After 2 h, the volume of water separated and accumulated on the surface (Vw) was recorded. The bleeding rate (BR, in %) was calculated as follows:BR = (Vw/V_0_) × 100%
where V_0_ represents the initial volume of the grout sample (200 mL) [29].

4.Mechanical Properties

Prismatic specimens measuring 40 mm × 40 mm × 160 mm were prepared and cured under standard conditions until the designated testing ages. The flexural and compressive strengths were determined using a 305F-2 microcomputer-controlled bending and compression integrated testing machine (Shenzhen Wance Testing Machine Co., Ltd., Shenzhen, China), in accordance with the Chinese National Standard GB/T 17671-2021 [30] (Methods of testing cements—Determination of Strength). The loading rates were set to 50 N/s and 2400 N/s for flexural and compressive tests, respectively. The reported strength values represent the arithmetic average of six individual specimens.

5.Scanning Electron Microscopy (SEM)

Fractured samples were collected at specific ages for microstructural morphology analysis. To arrest hydration, the fragments were immediately immersed in anhydrous ethanol for 48 h and subsequently dried in a vacuum oven at 60 °C. The dried samples were mounted on stubs, and a thin gold layer was sputter-coated onto the fracture surface to enhance conductivity. Microstructural observations were carried out using a scanning electron microscope operated at an accelerating voltage of 15 kV.

6.X-ray Diffraction (XRD)

For phase composition analysis, hydrated samples were ground into fine powder after hydration arrest and drying. The powder was sieved to obtain particles smaller than 80 μm. XRD analysis was performed using an X-ray diffractometer with Cu Kα radiation (λ = 1.5406 Å). The scanning range was set from 5° to 70° (2θ) with a step size of 0.02° and a scanning rate of 4°/min. The resulting diffraction patterns were analyzed using Jade software (version 6.0) by matching with standard powder diffraction files (PDFs) from the International Centre for Diffraction Data (ICDD).

7.Fourier-Transform Infrared Spectroscopy (FTIR)

Chemical structure and bonding information were obtained via FTIR spectroscopy. Dried sample powder was thoroughly mixed with spectroscopic-grade KBr at a mass ratio of approximately 1:100 and pressed into transparent pellets. Spectra were acquired in the mid-infrared region from 4000 to 400 cm^−1^ with a resolution of 4 cm^−1^, and each spectrum was averaged over 32 scans to improve the signal-to-noise ratio.

8.Mercury Intrusion Porosimetry (MIP)

Pore structure characteristics, including pore size distribution and total porosity, were determined using MIP. Dried samples of specified size were placed in a penetrometer. Mercury was intruded into the pores under applied pressure, which was gradually increased from 0.5 to 30,000 psi. The pore size distribution was derived from the intrusion data based on the Washburn equation, assuming a mercury surface tension of 485 mN/m and a contact angle of 130°.

## 3. Results and Discussion

### 3.1. Fresh State Properties

#### 3.1.1. Setting Time

This study systematically investigated the effects of nano-silica (NS) and graphene oxide (GO) on the setting times of cement paste at a fixed water-to-cement (w/c) ratio of 0.6. A constant dosage of 0.6 wt% polycarboxylate ether (PCE) superplasticizer was used in the NS series. As illustrated in Figure 2, the control group (without NS or GO) exhibited initial and final setting times of 958 min and 1084 min, respectively. The incorporation of NS markedly shortened both setting times in a dose-dependent manner. At NS dosages of 2 wt%, 3 wt%, and 4 wt%, the initial setting time decreased to 665 min, 592 min, and 547 min, while the final setting time was reduced to 795 min, 739 min, and 707 min, respectively. On average, each 1 wt% increase in NS content shortened the initial and final setting times by approximately 102.75 min and 94.25 min, respectively. Similarly, GO also exhibited a significant accelerating effect. With GO dosages of 0.01 wt%, 0.02 wt%, and 0.03 wt%, the initial setting times were recorded as 681 min, 664 min, and 654 min, and the final setting times as 739 min, 723 min, and 711 min, respectively. Notably, even a minimal GO dosage of 0.01 wt% considerably accelerated the setting process. Each 0.01 wt% increase in GO content reduced the initial and final setting times by approximately 26 min and 50 min, respectively. These results confirm that both NS and GO serve as effective setting accelerators, though likely through distinct mechanisms that warrant further microstructural investigation.

Figure 3 presents the setting times of the hybrid NS/GO groups. Comparative analysis revealed a clear synergistic effect between NS and GO within the composite system. When 0.01 wt% GO was introduced at constant NS dosages (compare mixes D2-01 vs. B2, D3-01 vs. B3, D4-01 vs. B4), the system exhibited a characteristic delay in initial setting but an acceleration in final setting. Conversely, the addition of 2 wt% NS at constant GO dosages (compare D2-01 vs. C01, D2-02 vs. C02, D2-03 vs. C03) resulted in significantly accelerated initial setting with only minor adjustments to final setting. This indicates that the high reactivity of NS effectively counteracted the potential retardation tendency of GO, ensuring rapid initiation of the setting reaction. These findings suggest that the NS/GO hybrid system modulates the cement setting process across different temporal stages rather than merely combining individual effects. While NS enhances reaction efficiency, GO improves the quality of structural formation. Together, they provide a critical foundation for enhancing the macroscopic properties of the system.

#### 3.1.2. Bleeding Rate and Fluidity

The effects of NS and GO on the bleeding rate and fluidity (flow spread) of the cement paste are shown in Figure 4 and Figure 5. The control group with 0.6 wt% PCE (NS baseline) showed a bleeding rate of 5.0% and a fluidity of 343.63 mm, while the group without PCE (GO baseline) exhibited a bleeding rate of 7.5% and a fluidity of 231.69 mm. PCE improves dispersion and fluidity through adsorption and steric hindrance, thereby enhancing stability. Increasing the NS dosage significantly improved stability. At NS dosages of 2 wt%, 3 wt%, and 4 wt%, the bleeding rate decreased to 2.25%, 2.0%, and 1.50%, respectively, demonstrating its strong anti-bleeding capacity. However, this improvement occurred at the expense of fluidity, which dropped to 239.70 mm, 221.46 mm, and 195.00 mm, respectively. On average, each 1 wt% increase in NS content reduced the bleeding rate by 0.875% and the fluidity by 37.16 mm, indicating that NS enhances stability but considerably increases flow resistance. GO incorporation also significantly improved stability, though its adverse effect on fluidity was much less pronounced than that of NS. At GO dosages of 0.01 wt%, 0.02 wt%, and 0.03 wt%, the bleeding rate decreased progressively from 5.25% to 2.50%, while fluidity only slightly decreased from 227.77 mm to 216.10 mm. Each 0.01 wt% increase in GO content reduced the bleeding rate by an average of 1.67% and the fluidity by only 5.2 mm, confirming that GO efficiently enhances stability at very low dosages while maintaining good fluidity.

The observed trends in fresh properties align well with the current understanding of cementitious material modification [31,32]. Extensive studies indicate that nanomaterials significantly influence cement hydration and rheological behavior due to their high specific surface area and surface effects [31,32]. The accelerating effect of NS primarily stems from its high pozzolanic reactivity, which enables rapid consumption of Ca(OH)_2_ to form additional C–S–H gel, while providing abundant nucleation sites for hydration products [33,34], thereby accelerating the dissolution and precipitation of tricalcium silicate (C_3_S). In contrast, GO relies more on its unique two-dimensional sheet structure and oxygen-containing functional groups to regulate the setting process by influencing ion migration (e.g., Ca^2+^), altering solution chemistry, and guiding the ordered crystallization of hydration products on its surface [35,36]. The systematic data obtained in this study not only corroborate these mechanisms but also, through hybrid experiments, reveal the competitive and synergistic interactions when both materials are present, offering a critical theoretical and experimental basis for the precise design of workability in cementitious materials.

### 3.2. Mechanical Properties

#### 3.2.1. Effect of NS and GO

As shown in Figure 6 and Figure 7, the effect of nano-silica (NS) content on the compressive and flexural strength of cement stone was investigated under a fixed water-cement ratio of 0.6 and a PCE dosage of 0.6 wt%. The results indicate that the addition of NS significantly improves both the ultimate flexural strength (UFS) and uniaxial compressive strength (UCS). However, this enhancement follows a non-linear trend: strength values peak at around 4 wt% NS, beyond which a “diminishing returns” behavior is observed. This non-monotonic response suggests that the influence of NS is not a simple function of concentration. The initial increase in strength at low to moderate NS contents (≤4 wt%) can be attributed to microstructural refinement driven by physicochemical interactions. In contrast, higher dosages (e.g., 5 wt%) likely induce nanoparticle agglomeration, which disrupts microstructure formation and consequently impairs macroscopic performance. Thus, determining the optimal NS dosage requires balancing its positive effects and potential detrimental impacts.

Figure 8 and Figure 9 present the strength results for pastes with different GO dosages (w/c = 0.6). The results demonstrate that even trace amounts of GO significantly enhanced strength, with a particularly pronounced improvement in compressive strength. As the GO dosage increased, both compressive and flexural strengths initially increased and then plateaued, indicating high efficiency of GO at low dosage ranges. The sensitivity of compressive strength to GO dosage suggests its substantial role in improving material compactness. The enhancement in flexural strength implies that GO also improved toughness and deformation resistance. This differential response indicates that GO likely influences compressive and flexural properties through distinct mechanisms.

#### 3.2.2. Effect of NS/GO Hybrids

Figure 10 and Figure 11 illustrate the effect of hybrid NS/GO incorporation on the compressive and flexural strengths of hardened cement paste. At a low NS dosage (2 wt%), the addition of 0.01 wt% GO significantly enhanced early-age strength: the 3-d flexural strength increased from 3.46 MPa to 4.84 MPa, and the 7-d compressive strength increased from 17.37 MPa to 23.81 MPa. Furthermore, the 28-d compressive strength also rose from 25.37 MPa to 29.61 MPa, indicating a clear improvement in later-age strength. However, as the NS dosage increased to 3 wt% and 4 wt%, further GO addition resulted in an overall decline in strength, suggesting a negative interaction between excessive NS and GO. At a low GO dosage (e.g., 0.01 wt%), the incorporation of 2 wt% NS markedly enhanced early-age strength and improved later-age strength to some extent. Nevertheless, as the GO dosage increased, the beneficial effect of NS gradually diminished and even adversely influenced later-age strength development.

The results reveal a clear interactive effect between NS and GO in the hybrid system. An appropriate ratio (e.g., 2 wt% NS with 0.01 wt% GO) resulted in a synergistic enhancement. However, excessive dosage of either component led to composite strengths lower than those achieved with individual components, underscoring the critical importance of optimizing the component ratio. The strength data suggest an optimal dosage range for NS/GO hybrids, beyond which further addition may impair performance.

The evolution of mechanical properties observed in this study aligns with numerous existing research conclusions, revealing common mechanisms in nanomaterial-modified cement. The peak enhancement effect of NS at around 4 wt% is consistent with the findings of Supit and Sha et al. [37], who reported that NS enhances early and later-age strength through pozzolanic reaction and micro-filling effects, but its efficiency reaches a saturation point; excessive addition leads to particle agglomeration and increased water demand, resulting in diminished strength gain or even performance deterioration. The significant enhancement in compressive and flexural strength by GO at very low dosages can be attributed to its two-dimensional sheet structure, which guides the morphology of hydration products and enhances material toughness [38]. A key finding of this study is the synergistic effect observed at low NS/GO dosages and the antagonistic effect upon excessive dosage of either component. This aligns with the research conclusions of Du and Pang [39] on nanomaterial hybrid systems, indicating that the dispersion, interaction, and influence on the hydration process of different nanomaterials in the matrix vary significantly. Precise optimization of their hybrid ratio is essential for leveraging synergistic effects and avoiding performance offset. Therefore, the effectiveness of NS/GO hybrids is highly dependent on the dosage ratio, and further microstructural analysis is needed to elucidate the specific mechanisms underlying their synergistic or antagonistic interactions.

### 3.3. Analysis of Hydration Products and Solidification Mechanisms in Grouting Materials

#### 3.3.1. SEM Analysis

Figure 12 presents representative scanning electron microscopy (SEM) images illustrating the microstructural morphology of 28-d cement pastes, highlighting the effects of different modification strategies. Figure 12 (A0), the unmodified cement, exhibits a typical loose and porous “coral-like” structure. Abundant hexagonal Portlandite (Ca(OH)_2_) crystals (red arrows) are loosely stacked, with substantial micron-scale pores and cracks (red circles) forming an interconnected network. This microstructure explains the material’s high permeability and relatively low mechanical strength.

Figure 12 (B4) shows that the incorporation of NS fundamentally altered the precipitation, nucleation, and growth of hydration products, resulting in a distinctly different microstructure. The field of view is dominated by a dense, continuous “sponge-like” C–S–H gel matrix. Large Ca(OH)_2_ crystals are substantially consumed and replaced by a tightly aggregated matrix comprising abundant nano-scale spherical/flocculent C–S–H gel particles (red arrows). These high-specific-surface-area nano-gel particles effectively fill primary pores and cracks and encapsulate unhydrated cement grains, thereby significantly refining the pore structure (note the markedly reduced pore size within the circles). This morphology underpins the superior performance of the B4 group and provides direct visual evidence of NS’s dual mechanisms: the “pozzolanic effect” consuming CH and the “micro-filler effect” optimizing porosity.

Figure 12 (B5) indicates microstructural degradation at an NS dosage of 5 wt%. The sample displays a highly porous and loose morphology composed of irregular plate-like crystals stacked loosely, forming abundant sheltered pores and interconnected cracks (red circles). Some spherical particles (red arrows) failed to fill pores effectively or form a continuous matrix, suggesting that excessive NS led to agglomeration, hindering dense C–S–H gel formation and ultimately causing microstructural deterioration and performance regression.

Figure 12 (C04) shows that incorporating 0.04 wt% GO significantly optimized the microstructure. Hydration products transformed into a dense monolithic structure formed by intensive accumulation and interlocking of irregular plate-like crystals. Compared to group A0, pores were effectively filled with much finer material, greatly enhancing structural continuity. The rough surface of the filling material in the lower-left corner suggests stronger interfacial bonding. Abundant C–S–H gel particles are attached to GO sheets, indicating that GO nanosheets provide nucleation sites through their “templating effect,” guide denser product arrangement, refine pores via the “micro-filler effect,” and enhance interfacial bond strength through “bridging” [40,41]. The synergy of these mechanisms underlies the excellent macroscopic performance of sample C04.

The comparison between Figure 12 (D2-01) and Figure 12 (D2-03) reveals a synergy-to-antagonism transition in the hybrid system. In D2-01, hydration products exhibit an interwoven, fluffy growth pattern. Plate-like and needle-like crystals cluster together, forming an open yet continuous structure with fine granular material, indicating favorable synergy: GO templates uniform growth, while NS optimizes pores through pozzolanic and micro-filling effects. In contrast, D2-03 is dominated by abundant regular, thin plate-like crystals stacked tightly in a “book-like” or fan-shaped arrangement, presenting a highly ordered but rigid laminated structure. This suggests excessive GO altered the hydration environment, dominated the process, inhibited C–S–H formation, and promoted lower-strength AFm phase formation. This structural change, combined with potential GO agglomeration, explains why D2-03′s compressive strength, though slightly higher than D2-01, regressed compared to the C03 sample (0.03 wt% GO alone). This provides direct evidence of a synergy–antagonism transition, emphasizing the necessity of precise nanomaterial dosage control.

#### 3.3.2. FTIR Analysis

Figure 13 presents the Fourier-transform infrared (FTIR) spectra of various samples at 28 days. The characteristic absorption band at ~1410 cm^−1^ is attributed to the asymmetric stretching vibration of the C–O bond in calcium carbonate (CaCO_3_). Its presence, along with features at ~870 cm^−1^ (C–O out-of-plane bending) and ~710 cm^−1^ (C–O in-plane bending), indicates varying degrees of carbonation (Ca(OH)_2_ + CO_2_→CaCO_3_ + H_2_O) during preparation or testing. The band near ~960 cm^−1^ corresponds to Si–O stretching in calcium silicate hydrate (C–S–H) gel, whose intensity and profile indicate the amount and structure of C–S–H formed. The broad feature around ~450 cm^−1^ likely arises from bending vibrations of SiO_4_^4−^ tetrahedra in unhydrated clinker (e.g., C_2_S, C_3_S) or quartz (SiO_2_). Detailed vibrational assignments are summarized in Table 5.

Comparative analysis of the FTIR spectra provides insights into the influence of different admixtures.

First, comparing A0 and B4 reveals that B4 (4 wt% NS) shows a significantly enhanced absorption at 960 cm^−1^, confirming that NS participated in the pozzolanic reaction (Ca(OH)_2_ + SiO_2_ + H_2_O→C–S–H), promoting more C–S–H gel formation [42], consistent with improved mechanical performance. The weak feature around 440 cm^−1^ (Si–O bending) in A0 becomes a distinct peak in B4, due to Si–O bonds from incorporated NS nanoparticles and/or evolved clinker phases. The similar intensities of CaCO_3_-related bands (1410 cm^−1^ and 870 cm^−1^) indicate that NS did not significantly alter carbonation.

Comparing B4 and B5 shows that excessive NS (5 wt%) adversely affected hydration. B5 exhibits reduced intensity at 960 cm^−1^, suggesting less C–S–H gel formation, likely due to particle agglomeration reducing reactive surface area, diminishing pozzolanic efficiency and interfering with normal hydration. This explains the macroscopic regression in B5.

Comparing A0 and C04, incorporating 0.04 wt% GO also markedly enhanced the C–S–H band intensity at 960 cm^−1^ (from 0.039 to 0.070), indicating promoted C–S–H formation due to GO’s “templating effect” and “nucleation site effect.” Its two-dimensional structure guides ordered growth and microstructural optimization. The CaCO_3_-related bands remain consistent, showing no significant influence on carbonation.

Finally, D2-01 and D2-03 exhibit comparable intensities across all characteristic bands, indicating similarities in hydration product composition and structure, consistent with their similar macroscopic mechanical performance.

Note: The absence of OH or H_2_O peaks in the FTIR spectra is attributed to the pre-drying treatment of the specimens prior to testing. The spectral range of 4000–2000 cm^−1^ was truncated as it showed no discernible peaks yet occupied significant space in the figure, thereby improving visual clarity while retaining all meaningful spectral features.

#### 3.3.3. XRD Analysis

Figure 14 presents the X-ray diffraction (XRD) patterns of various samples at 28 days: (a) stacked patterns, (b) waterfall plot, and (c) comparative portlandite (Ca(OH)_2_) characteristic peak intensity. Identified crystalline phases include quartz (SiO_2_), ettringite (Ca_6_Al_2_(SO_4_)_3_(OH)_12_·26H_2_O/AFt), portlandite (Ca(OH)_2_), dicalcium silicate (Ca_2_SiO_4_/C_2_S), and tricalcium silicate (Ca_3_SiO_5_/C_3_S), with C_2_S and C_3_S representing unhydrated clinker phases.

The pozzolanic effect of NS is directly confirmed. Compared to A0, B4 (4 wt% NS) shows significantly reduced intensity of the Ca(OH)_2_ peak at ~18° (2θ), indicating that reactive SiO_2_ in NS consumed calcium hydroxide to form additional amorphous C–S–H gel. This microstructural modification underpins the enhanced mechanical properties of B4.

An optimal NS dosage is evident. At 5 wt% NS (B5), the Ca(OH)_2_ peak reduction is less pronounced than in B4, suggesting agglomeration reduced specific surface area and reactivity, weakening the pozzolanic effect and causing performance regression.

The C04 sample (0.04 wt% GO) also shows a distinct decrease in Ca(OH)_2_ peak intensity. This supports GO’s physical toughening mechanisms (templating and bridging via its two-dimensional structure) and implies that its surface functional groups may catalyze hydration, promoting CH consumption and product formation, enhancing performance through a synergistic chemical pathway.

Furthermore, after the addition of NS/GO, the intensities of the crystalline diffraction peaks of both AFt and AFm were weakened. This might be because NS/GO altered the hydration path, causing the system to tend to form amorphous or gel-like calcium aluminate hydrates (such as the complex of Al(OH)_3_ gel and C–S–H gel). This type of substance is difficult to be detected in the XRD pattern.

Critically, the GO/NS hybrid system exhibits a clear synergy–antagonism transition. The low-dosage hybrid (D2-01) shows Ca(OH)_2_ peak intensity significantly lower than A0, indicating GO improved NS dispersion and reactivity, yielding synergistic enhancement. In contrast, the high-dosage hybrid (D2-03) shows less Ca(OH)_2_ consumption than D2-01, revealing that excessive GO induced hetero-agglomeration, inhibiting NS activity and turning synergy into antagonism, leading to performance degradation. This provides a theoretical basis for rational design and dosage control of multi-nanomaterial systems in cementitious matrices.

#### 3.3.4. MIP Analysis

Figure 15 and Figure 16 present the pore size distribution and porosity characteristics at 28 days, measured by mercury intrusion porosimetry (MIP). Based on established criteria [43], pores were categorized into: micropores (MICP, <10 nm), medium capillary pores (MCP, 10–50 nm), large capillary pores (LCP, 50–1000 nm), and macropores (MACP, >1 μm).

Analysis reveals that NS and GO composite modification exhibits complex behavior with synergistic and antagonistic effects. NS showed significant pozzolanic activity and micro-filling effect. B4 (4 wt% NS) had markedly reduced total pore volume versus A0. Macropore and medium capillary pore volumes decreased significantly, while micropore and medium capillary pore proportions increased, indicating NS consumes Ca(OH)_2_ to form C–S–H gel clogging larger pores and acts as an ultra-fine filler, refining pores and reducing porosity. This dual mechanism explains the mechanical enhancement.

An optimal NS dosage limit was confirmed. At 5 wt% NS (B5), total pore volume rebounded, and the differential curve showed a higher macropore peak, suggesting agglomeration created defect centers introducing harmful large pores, deteriorating the pore structure.

GO exhibited a complementary mechanism. C04 (0.04 wt% GO) had slightly lower total pore volume than A0, with uniform reduction across all ranges and a flatter differential curve, indicating GO functions through templating and bridging, guiding a more uniform and dense microstructure, holistically optimizing pore distribution.

Notably, D2-03 had slightly lower total porosity than D2-01 and a relatively lower proportion of harmful macropores, explaining its marginally superior mechanical performance. Although both D2-01 and D2-03 had lower total porosity than A0, their macropore proportions were significantly higher, indicating GO/NS synergy might adversely affect pore size distribution. However, in D2-03, smaller pores were more refined, and GO’s bridging effect strengthened the pore skeleton, compensating for increased macropores, yielding overall strength improvement.

## 4. Conclusions

This study systematically investigated the individual and synergistic effects of nano-silica (NS) and graphene oxide (GO) on the macroscopic properties and microstructure of cement-based grouting materials. The results provide both theoretical insights and experimental support for the precise design and application of nanomaterials in cementitious systems. The main conclusions are as follows:Fresh State Properties: Both NS and GO significantly accelerated the setting time of cement paste and effectively reduced the bleeding rate. However, NS had a more detrimental impact on fluidity compared to GO. The hybrid system further shortened the setting process and improved stability, albeit at the cost of a more substantial reduction in fluidity.Mechanical Properties: Both NS and GO significantly enhanced the compressive and flexural strength of cement paste at optimal dosages. The optimal dosage of NS was 4 wt%, while GO exhibited notable reinforcing effects within the range of 0.01–0.03 wt%. The hybrid system displayed a distinct transition from synergistic to antagonistic effects: a low-dosage combination (e.g., 2 wt% NS + 0.01 wt% GO) synergistically improved mechanical properties, increasing flexural and compressive strength by 13.5% and 45.5%, respectively, compared to the reference group. However, excessive dosage of either component (e.g., 5 wt% NS or 0.03 wt% GO in combination) resulted in reduced reinforcement efficiency, with performance even inferior to that of samples incorporating a single nanomaterial.Microstructure:NS enhances the matrix primarily through the pozzolanic reaction—consuming Ca(OH)_2_ to form additional C–S–H gel—and a nano-filling effect that optimizes the pore structure. In contrast, GO improves performance via a templating effect and by providing nucleation sites that promote dense deposition of hydration products, along with a bridging effect that enhances overall integrity.Excessive NS led to agglomeration, increasing the volume of harmful pores. Excessive GO resulted in abnormal morphology of hydration products (e.g., stacked plate-like AFm phases), inhibiting the formation of C–S–H gel.In the hybrid system, GO improved the dispersion of NS and enhanced its pozzolanic effect. However, excessive GO induced hetero-agglomeration, which weakened the activity of NS and caused a transition from synergy to antagonism.Pore Structure: NS mainly reduced medium and large capillary pores (>50 nm) and increased the proportion of fine pores (<50 nm), significantly decreasing total porosity. GO uniformly reduced pores across all size ranges, resulting in a more homogeneous pore size distribution. Although the hybrid system reduced total porosity, it also increased the proportion of large pores, the long-term durability implications of which require further investigation.This study demonstrates that both NS and GO offer significant potential for modifying cement-based grouting materials, but their dosages must be strictly optimized. It is recommended that the NS dosage should not exceed 4 wt%, and the GO dosage should remain below 0.03 wt%. For hybrid systems, a low-to-medium dosage combination (e.g., 2 wt% NS + 0.01 wt% GO) is recommended to avoid agglomeration and antagonistic effects, thereby achieving synergistic performance enhancement.

## Figures and Tables

**Figure 1 materials-18-04820-f001:**
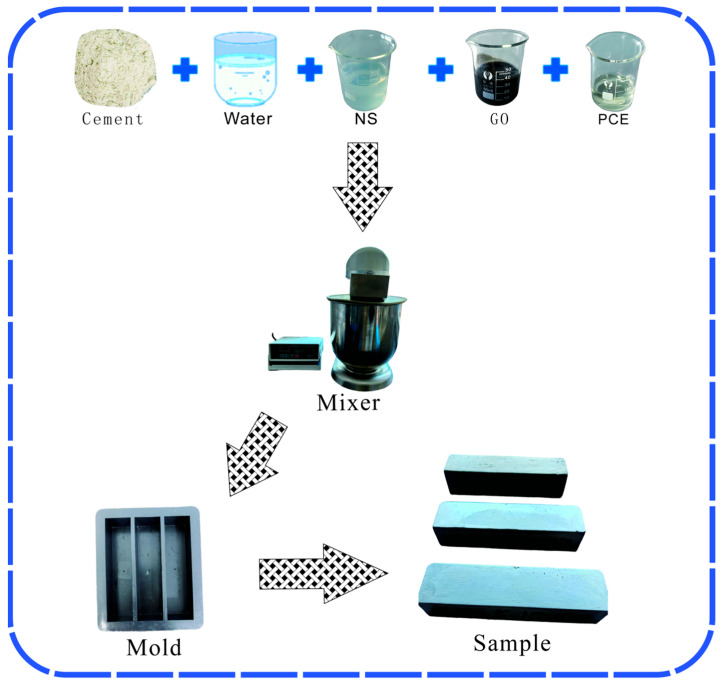
Sample preparation.

**Figure 2 materials-18-04820-f002:**
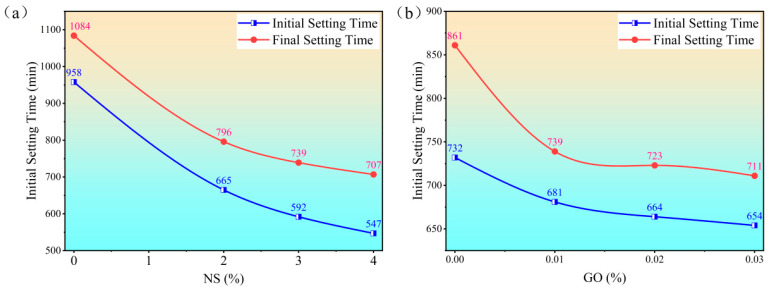
Effect of (**a**) NS and (**b**) GO dosage on the setting time of cement paste.

**Figure 3 materials-18-04820-f003:**
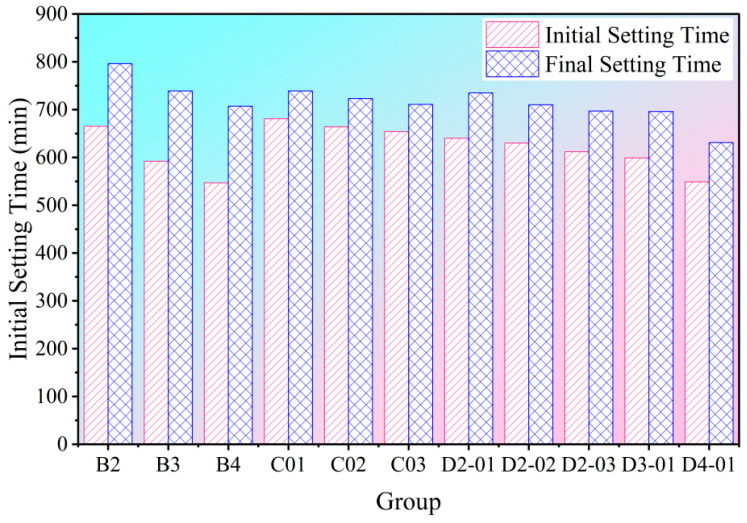
Setting times of each NS/GO group.

**Figure 4 materials-18-04820-f004:**
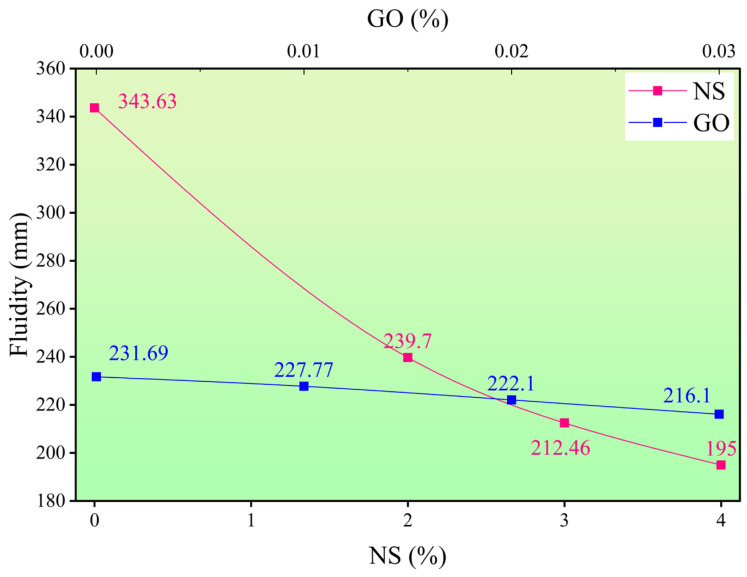
Effect of nano-silica (NS) and graphene oxide (GO) dosage on the fluidity (flow spread) of cement paste.

**Figure 5 materials-18-04820-f005:**
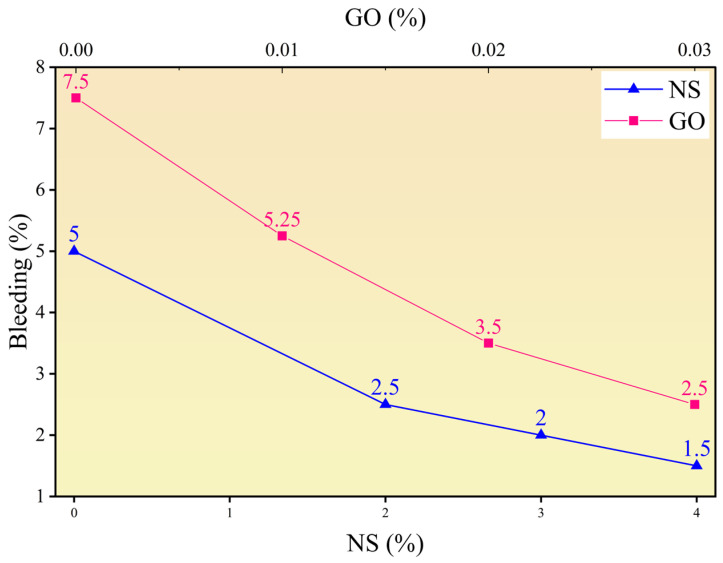
Effect of nano-silica (NS) and graphene oxide (GO) dosage on the bleeding rate of cement paste.

**Figure 6 materials-18-04820-f006:**
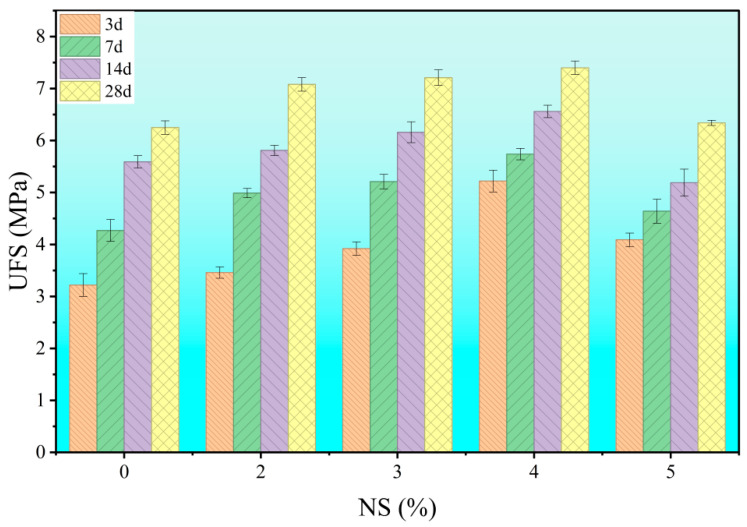
Effect of nano-silica (NS) dosage on the flexural strength of hardened cement paste.

**Figure 7 materials-18-04820-f007:**
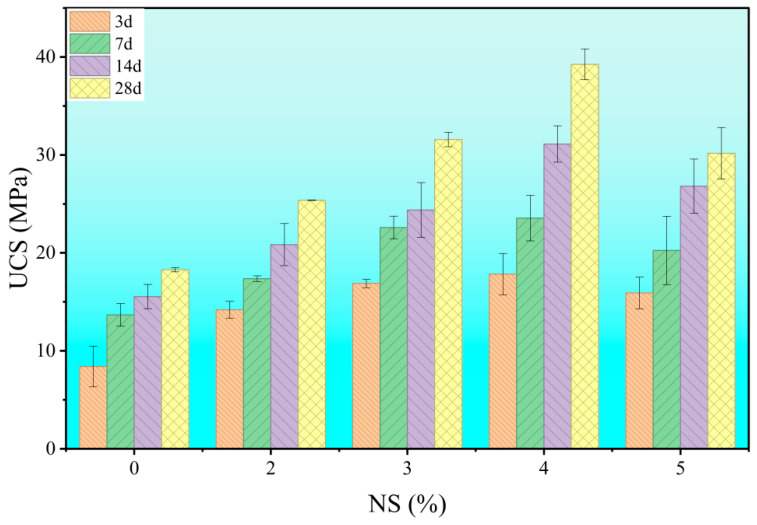
Effect of nano-silica (NS) dosage on the compressive strength of hardened cement paste.

**Figure 8 materials-18-04820-f008:**
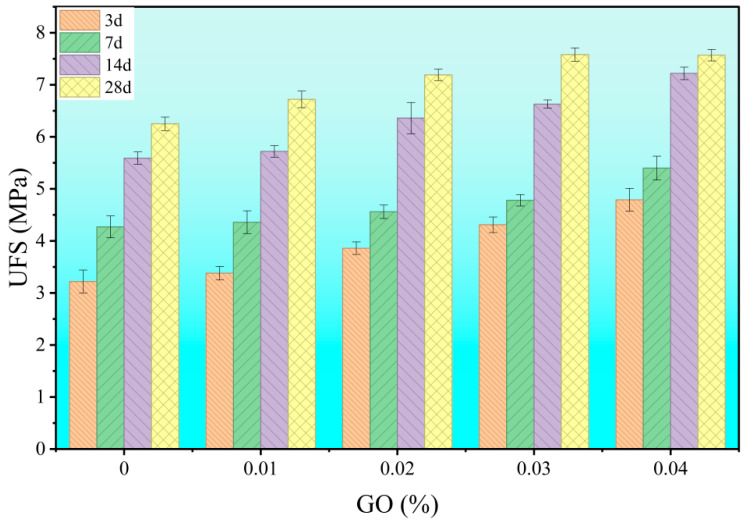
Effect of graphene oxide (GO) dosage on the flexural strength of hardened cement paste.

**Figure 9 materials-18-04820-f009:**
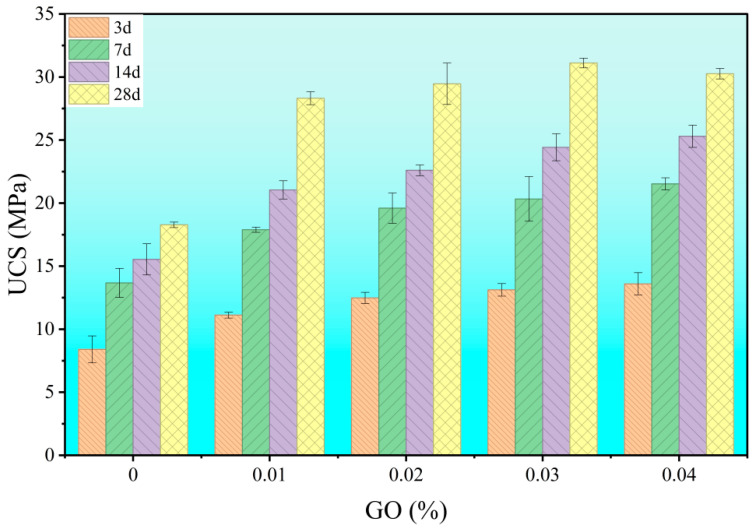
Effect of graphene oxide (GO) dosage on the compressive strength of hardened cement paste.

**Figure 10 materials-18-04820-f010:**
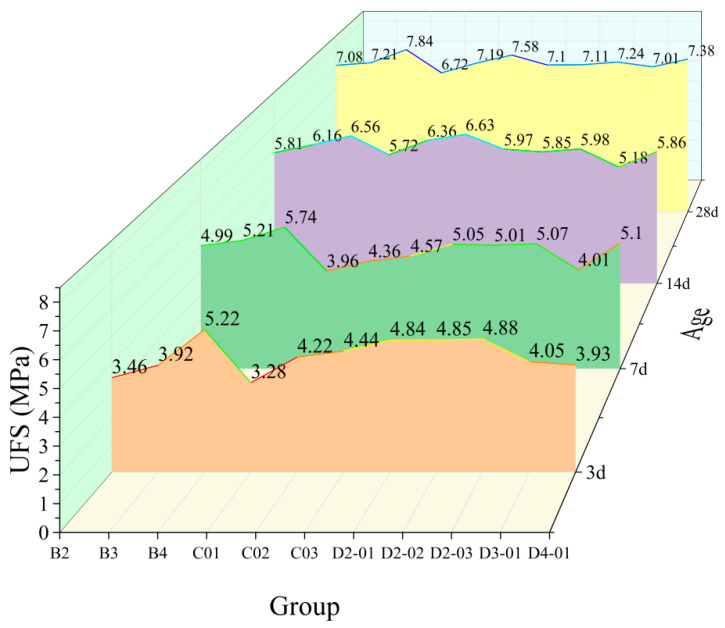
Effect of NS/GO hybrids on the flexural strength of hardened cement paste.

**Figure 11 materials-18-04820-f011:**
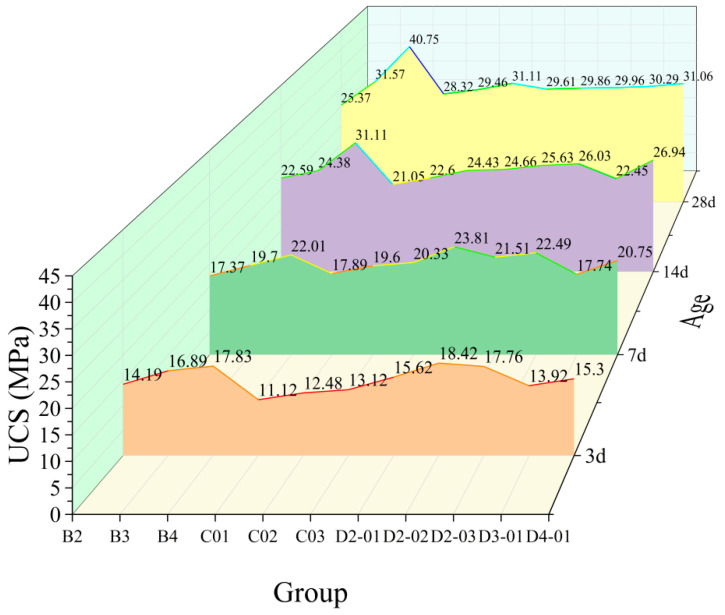
Effect of NS/GO hybrids on the compressive strength of hardened cement paste.

**Figure 12 materials-18-04820-f012:**
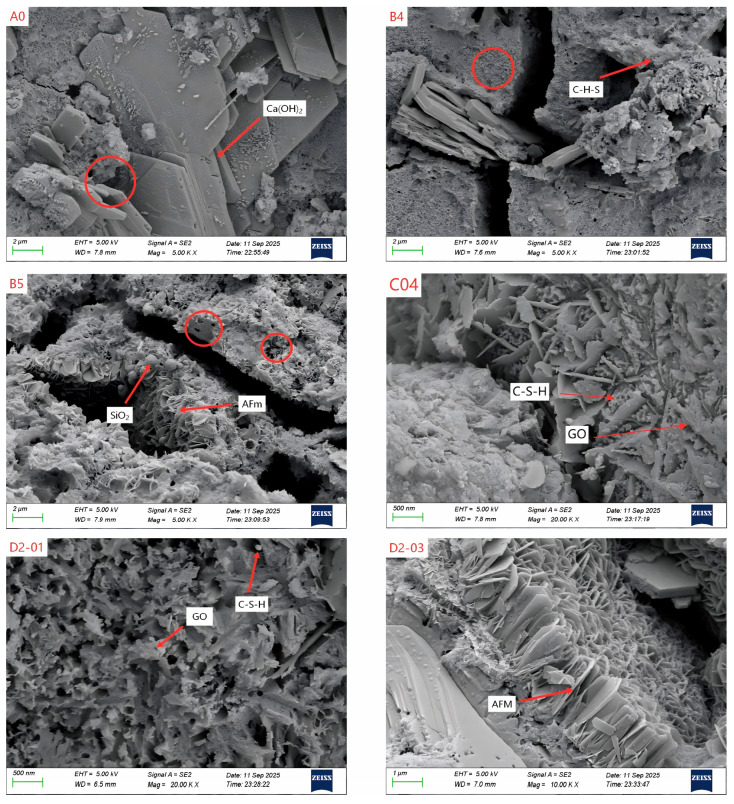
SEM images of different groups of samples.

**Figure 13 materials-18-04820-f013:**
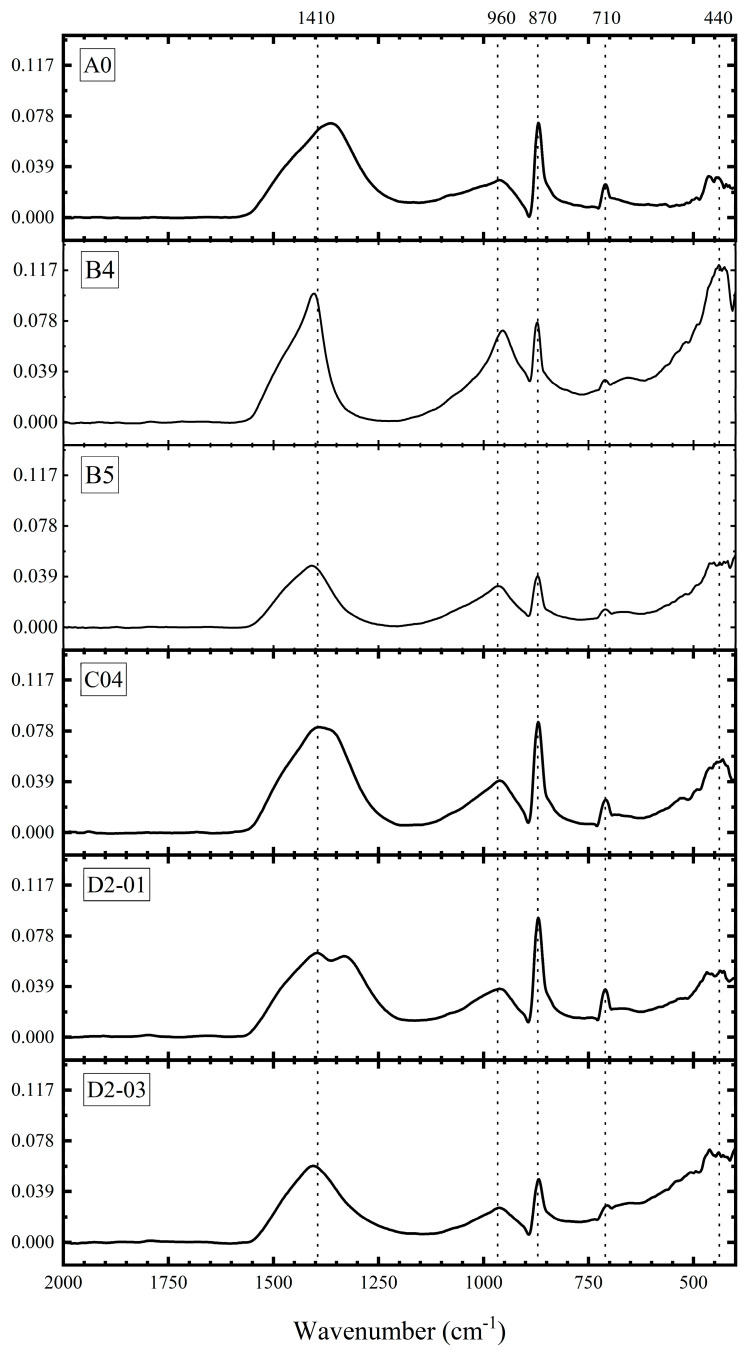
FTIR spectra of different groups of samples.

**Figure 14 materials-18-04820-f014:**
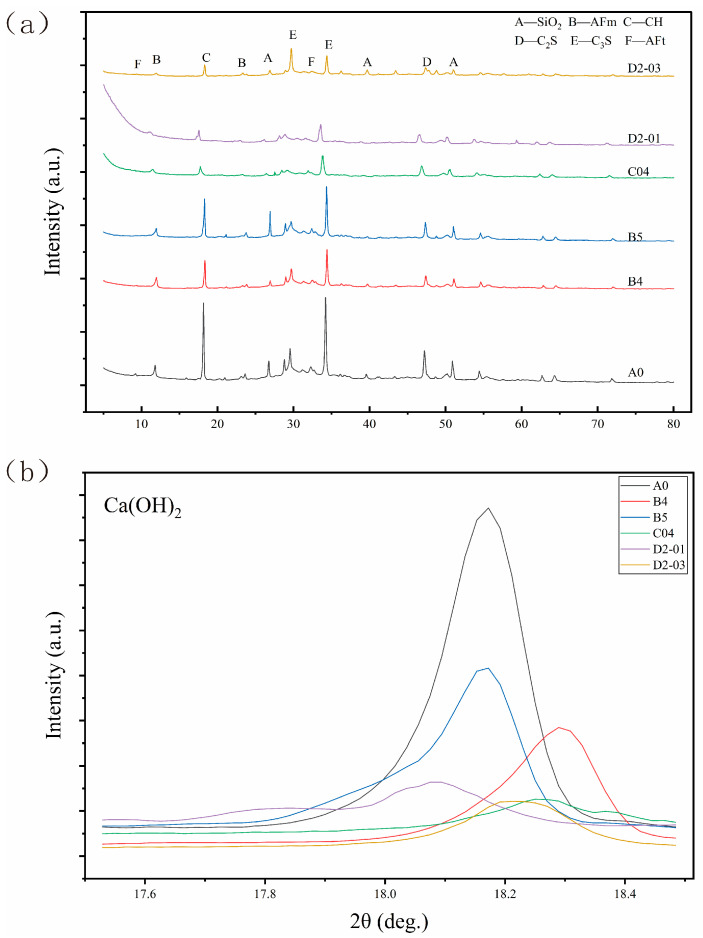
XRD patterns of different groups of samples: (**a**) stacked patterns (**b**) comparison of Ca(OH)_2_ characteristic peak intensity.

**Figure 15 materials-18-04820-f015:**
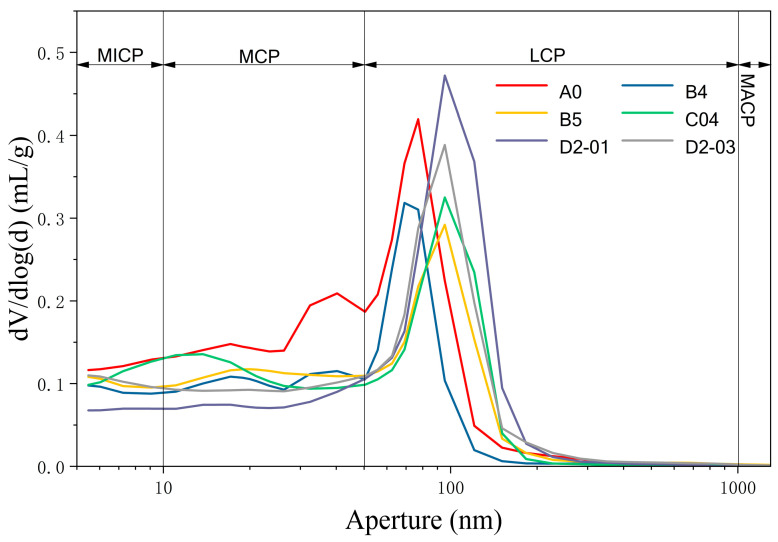
Pore size distribution of different sample groups.

**Figure 16 materials-18-04820-f016:**
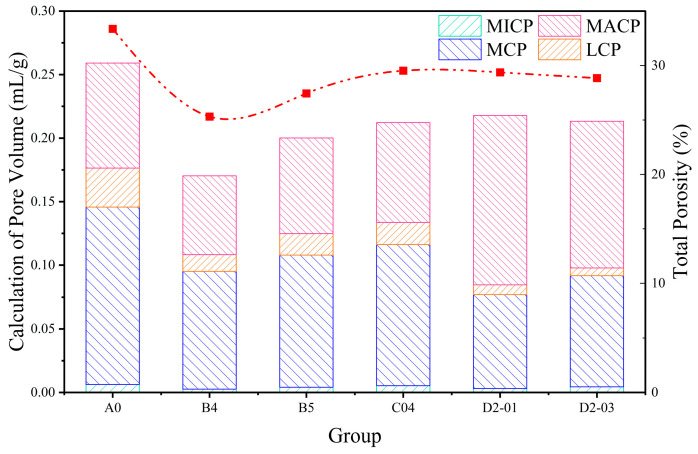
Porosity characteristics of different sample groups.

**Table 1 materials-18-04820-t001:** Chemical Composition and Phase Composition of Portland Cement (mass %).

Compound	Content (%)	Phase Name	Approximate Content
CaO	54.36%	C_3_S	45–60%
SiO_2_	22.31%	C_2_S	15–30%
Al_2_O_3_	9.76%	C_3_A	6–12%
SO_3_	3.16%	C_4_AF	6–8%
Fe_2_O_3_	3.13%	Gypsum	3–5%
K_2_O	1.03%	Additives	5–20%
MgO	1.01%		
TiO_2_	0.43%		

**Table 2 materials-18-04820-t002:** Technical specifications of the nano-silica (NS) solution.

Property	Value
SiO_2_ content	40.1 wt%
Na_2_O	0.22 wt%
pH value	9.6
Average particle size	10 nm
Density(at 25 °C)	1.295 g/cm^3^
Appearance	Semi-translucent, milky-white colloid

**Table 3 materials-18-04820-t003:** Physicochemical properties of the graphene oxide (GO) dispersion.

Property	Value
Number of layers	Single-layer
Single-layer ratio	>99.9%
Thickness	~1 nm
Flake diameter	10–50 μm
Oxidation degree	~36%
Specific surface area	50–150 m^2^/g

**Table 4 materials-18-04820-t004:** Mix proportions of the cement-based grouting materials.

Group	w/c	Cement	SP (wt%)	NS (wt%)	GO (wt%)
A0	0.6	100%	0.6	/	/
B2	0.6	100%	0.6	2	/
B3	0.6	100%	0.6	3	/
B4	0.6	100%	0.6	4	/
B5	0.6	100%	0.6	5	/
C01	0.6	100%	/	/	0.01
C02	0.6	100%	/	/	0.02
C03	0.6	100%	/	/	0.03
C04	0.6	100%	/	/	0.04
D2-01	0.6	100%	0.6	2	0.01
D2-02	0.6	100%	0.6%	2	0.02
D2-03	0.6	100%	0.6	2	0.03
D3-01	0.6	100%	0.6	3	0.01
D4-01	0.6	100%	0.6	4	0.01

Note: SP = Superplasticizer; wt% = percentage by mass of cement; The “water” in the water-cement ratio calculation refers to the total amount of water used for mixing, which includes: the added water, the water contained in the nano-silica sol, the water in the graphene oxide dispersion, and the water contained in the water-reducing agent solution.

**Table 5 materials-18-04820-t005:** Assignment of major FTIR absorption bands observed in the cement pastes.

Wavenumber (cm^−1^)	Vibration Assignment	Compound/Structure Indicated
~1410	C–O asymmetric stretching	Calcium carbonate (CaCO_3_)
~960	Si–O stretching	Calcium silicate hydrate (C–S–H) gel
~870–880	C–O out-of-plane bending	Calcium carbonate (CaCO_3_)
~710	C–O in-plane bending	Calcium carbonate (CaCO_3_)
~430–450	Si–O bending	SiO_4_^4−^ tetrahedron (in C_2_S, C_3_S, or quartz)

## Data Availability

The original contributions presented in this study are included in this article. Further inquiries can be directed to the corresponding author.

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
