# Peer review of "Cement-Based Grouting Materials Modified with GO/NS Hybrids"

_materials, 2025, doi:10.3390/ma18214820_

Round 1
Reviewer 1 Report
Comments and Suggestions for Authors
Dear Author,
I really appreciate your manuscript. You did a scientific and well supported analysis to evaluate the effects of GO's and NS's on cement-based materials. I do not have specific comments.
Just check figure 4 and 5; there may be a mistake in the legend. was NS substituted automatically by your graph software with fluidity and bleeding?
Congrats on your work
Author Response
Comments 1: Just check figure 4 and 5; there may be a mistake in the legend. was NS substituted automatically by your graph software with fluidity and bleeding?
Response 1: Thank you for bringing this to our attention. You are correct that a text replacement error occurred in the legends of Figures 4 and 5 during the export process. The issue has been corrected in the revised version (please see the updated Figures 4 and 5 on page 8 of the manuscript). We sincerely apologize for this oversight and have thoroughly verified all figure labels in the final version to ensure accuracy.

Reviewer 2 Report
Comments and Suggestions for Authors
Big work was done by authors (16 Figures).
I have several comments.
- The title sounds too much pretentious. In fact "Multi-Scale Synergistic Mechanisms" was not clearly described. I would suggest more simple title like "Cement-Based Grouting Materials Modified with GO/NS Hybrids".
- Starting raw materials presented, in my opinion, not correctly. How can chemical composition of Portland cement can help with understanding of the processes of cement stone formation? Phase composition have to be shown. And description of component "nanosilica" is not acceptable. pH=9,6 give me sign that you probably used water solution of Na2O.nSiO2. This starting material according description contained 40.1 wt% of SiO2. It is a secret what was other 59.9wt%. Water? What then with water/cement in table 4?
- I think that after data about setting time and other data in Figures 1-5 XRD of prepared samples have to be presented. But XRD was presented in the end of the article. And presented XRD (Figure 14) can not be accepted as correct data. We can see at the Figure 14 minerals of Portland cement C2S or C3S as if there were no interaction with water. According my understanding of cement stone formation XRD data have to show hydrated phases after solidification of cement mass. This question have to be answered with description of chemical transformation during cement stone formation. Otherwise repetition during text of article of word combination "Synergistic Mechanisms" has no sense.
- Names of Y axes at Figures 6-11 are not understandable.
- Choice of samples in Figure 12 (SEM) is not understandable, also as circles at those photos with different magnification (no opportunity to compare microstructure of different samples).
- FTIR analysis in part Materials and methods declare 4000 to 400 cm⁻¹ interval for implementation. In Figure 13 (FTIR spectra of different groups of samples) data presented in interval 2000-400. So no one can see any signs of OH or H2O.
- Crystalline phases ettringite (AFt), dicalcium silicate (C₂S), and tricalcium silicate (C₃S), with C₂S and C₃S representing without formulas.
- part Conclusion in this article has name "Discussion".

Author Response
We greatly appreciate the reviewers' insightful comments and have revised the manuscript accordingly. Below are our point-by-point responses:
Comments 1: The title sounds too much pretentious. In fact "Multi-Scale Synergistic Mechanisms" was not clearly described. I would suggest more simple title like "Cement-Based Grouting Materials Modified with GO/NS Hybrids".
Response 1: We sincerely thank the reviewer for this constructive suggestion. We agree that the original title may have overstated the mechanistic insights. Accordingly, we have revised the title to: "Cement-Based Grouting Materials Modified with GO/NS Hybrids" as recommended.
Comments 2: Starting raw materials presented, in my opinion, not correctly. How can chemical composition of Portland cement can help with understanding of the processes of cement stone formation? Phase composition have to be shown. And description of component "nanosilica" is not acceptable. pH=9,6 give me sign that you probably used water solution of Na2O.nSiO2. This starting material according description contained 40.1 wt% of SiO2. It is a secret what was other 59.9wt%. Water? What then with water/cement in table 4?
Response 2: We appreciate the reviewer’s thorough evaluation. In response, we have supplemented the table on Portland cement with its phase composition in addition to the chemical composition. Regarding the nano-silica solution, the reviewer is correct: it is an aqueous suspension stabilized with Na₂O, and the remaining content is indeed water. This clarification has been added to Table 2. Furthermore, all water introduced via solutions (including the nano-silica suspension) was accounted for in the water-to-cement ratio, which is now explicitly stated in the notes of Table 4.
Comments 3: I think that after data about setting time and other data in Figures 1-5 XRD of prepared samples have to be presented. But XRD was presented in the end of the article. And presented XRD (Figure 14) can not be accepted as correct data. We can see at the Figure 14 minerals of Portland cement C2S or C3S as if there were no interaction with water. According my understanding of cement stone formation XRD data have to show hydrated phases after solidification of cement mass. This question have to be answered with description of chemical transformation during cement stone formation. Otherwise repetition during text of article of word combination "Synergistic Mechanisms" has no sense.
Response 3: We are grateful for this insightful comment. The XRD analysis was placed later in the manuscript to correlate not only with the setting time and early properties (Figs. 1–5) but also with the mechanical performance (Figs. 6–11). In accordance with the suggestion, we have added labeling of key crystalline phases (including hydrated products) in Figure 14 and expanded the discussion regarding the hydration process. The presence of unhydrated C₂S and C₃S is typical at this curing age. Notably, their peak intensities decreased in samples with GO/NS, suggesting promoted hydration, which we have further emphasized in the revised text.
Comments 4: Names of Y axes at Figures 6-11 are not understandable.
Response 4: Thank you for pointing this out. We have updated the Y-axis labels in Figs. 6–11 to clearly indicate Ultimate Flexural Strength (UFS) and Ultimate Compressive Strength (UCS), with full terms defined in the Abbreviations section and subsection 3.2.
Comments 5: Choice of samples in Figure 12 (SEM) is not understandable, also as circles at those photos with different magnification (no opportunity to compare microstructure of different samples).
Response 5: We appreciate this comment. The circled areas in the SEM images (Fig. 12) now use the same magnification to facilitate direct microstructural comparison across samples.
Comments 6: FTIR analysis in part Materials and methods declare 4000 to 400 cm⁻¹ interval for implementation. In Figure 13 (FTIR spectra of different groups of samples) data presented in interval 2000-400. So no one can see any signs of OH or H2O.
Response 6: We thank the reviewer for highlighting this issue. The samples were dried before FTIR analysis, which explains the absence of OH/H₂O peaks. The spectral range was truncated (4000–2000 cm⁻¹ omitted) because no significant peaks appeared in that region, and displaying it reduced the clarity of meaningful bands. We have added a note in the caption to clarify this rationale.
Comments 7: Crystalline phases ettringite (AFt), dicalcium silicate (C₂S), and tricalcium silicate (C₃S), with C₂S and C₃S representing without formulas.
Response 7: We thank the reviewer for the careful reading. The crystalline phases are now presented with their chemical formulas: ettringite (Ca₆Al₂(SO₄)₃(OH)₁₂·26H₂O / AFt), dicalcium silicate (Ca₂SiO₄ / C₂S), and tricalcium silicate (Ca₃SiO₅ / C₃S) in subsection 3.3.3.
Comments 8: part Conclusion in this article has name "Discussion".
Response 8: We apologize for this oversight. The “Discussion” heading has been corrected to “Conclusions”, and we have incorporated key quantitative findings summarizing the major achievements of the study.

Reviewer 3 Report
Comments and Suggestions for Authors
Reviewer's report on the paper titled “Performance Evolution and Multi-Scale Synergistic Mechanisms of Cement-Based Grouting Materials Modified with GO/NS Hybrids” for the MDPI Journal Materials
The paper “Performance Evolution and Multi-Scale Synergistic Mechanisms of Cement-Based Grouting Materials Modified with GO/NS Hybrids” is an original contribution, which corresponds to the scope of MDPI Journal Materials.
The synergistic effects of graphene oxide (GO) and nano-silica (NS) on cement-based grouting materials to enhance performance through microstructural refinement was investigated in the current study. The macroscopic properties of cement-based grouting materials such as setting time, fluidity, bleeding rate, and mechanical strength, were evaluated alongside multi-scale microstructural characterization using SEM, XRD, MIP, and FTIR. The results demonstrate that both nano-silica and graphene oxide reduced setting time and bleeding rate while improving mechanical properties though nano-silica more significantly decreased fluidity.
The following improvements/complications should be done to made the paper more clear for its understanding:
- Abstract of the current study should be expanded and completed by the major numerical results, illustrating the most significant observations, obtained in course of the current study.
- Sub-chapter 2.2. “Sample Preparation” should be completed by the geometric parameters of the specimens, shown in the Figure 1.
- Determination of mechanical properties of the specimens, described in the sub-chapter 2.3. “Testing Methods”, should be described in the more details. Standards used for the procedure specifying so as the testing equipment should be mentioned and described.
- Authors are advised to check the colored designations in the Figure 16. The designations include four colors, but the diagrams in the figure include only three once.
- The paragraphs numbering in the Chapter 4 “Discussion” is look like a strange, so as is started form number 13 and finished by the number 17. So, it should be improved.
- The paper should be supplied by the chapter “Conclusions”, which should summarize the most significant achievements, obtained in course of the current study. The major numerical results were obtained should be added in the chapter.
Author Response
We greatly appreciate the reviewers' insightful comments and have revised the manuscript accordingly. Below are our point-by-point responses:
Comments 1: Abstract of the current study should be expanded and completed by the major numerical results, illustrating the most significant observations, obtained in course of the current study.
Response 1:Thank you for your suggestion. We have expanded the abstract to include key quantitative results—such as a 13.5% increase in flexural strength and a 45.5% improvement in compressive strength under optimal hybrid dosages—along with the central trends observed in this study. These additions more clearly highlight the significant findings of our research.
Comments 2: Sub-chapter 2.2. “Sample Preparation” should be completed by the geometric parameters of the specimens, shown in the Figure 1.
Response 2:Thank you for pointing this out. We have added a detailed description of the geometric parameters (40 mm × 40 mm × 160 mm) of the specimens shown in Figure 1 to Subsection 2.2 "Sample Preparation" to enhance methodological clarity.
Comments 3: Determination of mechanical properties of the specimens, described in the sub-chapter 2.3.“Testing Methods”, should be described in the more details. Standards used for the procedure specifying so as the testing equipment should be mentioned and described.
Response 3:We appreciate your comment. Section 2.3 "Testing Methods" has been updated with a thorough description of the mechanical testing procedures, including reference to the testing standard (GB/T 17671-2021) and detailed information about the testing equipment (Model: 305F-2 Combined Bend-Compression Testing Machine; Manufacturer: Shenzhen Wance Testing Machine Co., Ltd.) to ensure reproducibility.
Comments 4: Authors are advised to check the colored designations in the Figure 16.The designations include four colors, but the diagrams in the figure include only three once.
Response 4:Thank you for your feedback. We have reviewed Figure 16 and confirmed that the light color of the "MICP" data series—due to its small values and proximity to the x-axis—made it difficult to distinguish. This has been corrected using a more visually prominent color to ensure clear representation of all data series.
Comments 5: The paragraphs numbering in the Chapter 4 “Discussion” is look like a strange, so as is started form number 13 and finished by the number 17. So, it should be improved.
Response 5:We apologize for the oversight. The anomalous paragraph numbering in Chapter 4 ("Discussion") resulted from formatting inconsistencies during editing. We have renumbered all paragraphs to ensure sequential and logical coherence throughout the section.
Comments 6: The paper should be supplied by the chapter “Conclusions”, which should summarize the most significant achievements, obtained in course of the current study. The major numerical results were obtained should be added in the chapter.
Response 6:Thank you for highlighting this issue. We have corrected the erroneous labeling of the "Conclusions" section (previously mislabeled as "Discussion") and incorporated the major quantitative results—such as mechanical performance metrics under varying NS/GO ratios—along with central conclusions derived from this study.

Round 2
Reviewer 2 Report
Comments and Suggestions for Authors
Almost all my comments were answered.
Small comments additionally:
- Portland cement (P.O 42.5) means that it has compressive strength 42,5 MPa after 28 days. Yes? Probably you can give comment about this fact and compressive strength of sample you prepared.
- Having question about what is nano-silica is I would like to find the description of the nano-silica. But I did not find the company. Probably there is no site in English. You used "the nano-silica (NS) solution". SiO2 is not soluble. And Na2O can interact with water. Name nano-silica is beautiful, but can lead to the misunderstanding of chemical composition of the component. Could you please clarify?
- Table 4. The naming of samples is "not-saying". But you can leave the labeling as it is.
- Figure 3. Figure 10. Figure 11. What about A0?
- Page 17. Please check the sentence "Identified crystalline phases include quartz (SiO₂), ettringite...." It seems there are extra letters.
- And what about densities of the created samples with non-saying labeling (A, B, C, D)?

Author Response
Comments 1: Portland cement (P.O 42.5) means that it has compressive strength 42,5 MPa after 28 days. Yes? Probably you can give comment about this fact and compressive strength of sample you prepared.
Response 1: Thank you for this comment. The designation “P. O 42.5” refers to Portland Ordinary cement with a minimum compressive strength of 42.5 MPa at 28 days, as specified in the Chinese national standard GB/T 17671-2021.(P stands for Portland Cemen, O stands for Ordinary.) This strength is measured under standardized testing conditions, with a mixture ratio of cement : standard sand : water = 1 : 3 : 0.5.
Comments 2: Having question about what is nano-silica is I would like to find the description of the nano-silica. But I did not find the company. Probably there is no site in English. You used "the nano-silica (NS) solution". SiO2 is not soluble. And Na2O can interact with water. Name nano-silica is beautiful, but can lead to the misunderstanding of chemical composition of the component. Could you please clarify?
Response 2: Thank you for this thoughtful comment and for pointing out the potential ambiguity in the naming of the material. You are absolutely right to seek clarification, and we appreciate the opportunity to provide a more detailed explanation.
The term "nano-silica (NS) solution" indeed requires further specification to avoid misunderstanding. As the reviewer rightly noted, SiO2 itself is not soluble in water, and Na2O can undergo hydrolysis. The material used in our study was, in fact, an aqueous colloidal suspension of nano-silica, often commercially described as "sodium silicate stabilized nano-silica sol."
Here is a detailed breakdown of its composition:
Solid Phase: Amorphous Silicon Dioxide (SiO2)
Dispersing Medium: Deionized Water.
Stabilizing Agent: A small amount of Sodium Oxide (Na₂O, typically 0.3-0.5% by weight) is used to maintain the stability of the colloidal suspension by providing alkaline conditions (pH ~9.5-10.5), which prevent aggregation and ensure a uniform dispersion.
Therefore, the description "40.1 wt% SiO2" refers to the solid content of the suspension. The remaining mass is primarily water, with a minor fraction of the stabilizer (Na2O).
Comments 3: Table 4. The naming of samples is "not-saying". But you can leave the labeling as it is.
Response 3: We thank the reviewer for this valuable feedback. We agree that the sample labeling convention could be more intuitive. In the current system: A0 denotes the reference group without admixtures; BX represents groups with nano-silica (NS) content of X%; C0X indicates groups with graphene oxide (GO) content of 0.0X%; and DX-0Y corresponds to hybrid groups with X% NS and 0.0Y% GO. We truly appreciate the suggestion and will adopt a clearer and more informative naming strategy in future studies.
Comments 4: Figure 3. Figure 10. Figure 11. What about A0?
Response 4: Thank you for raising this question. The A0 group data were not included in Figures 3, 10, and 11 for two primary reasons: first, the results of the A0 reference group have already been clearly presented in the "0%" content column in Figures 2 and 6–9; second, these three figures specifically aim to highlight the synergistic effects of hybrid NS/GO incorporation (Group D), thus focusing on comparing the performance between the hybrid group (D-series) and the single-admixture groups (B and C-series).
Comments 5: Page 17. Please check the sentence "Identified crystalline phases include quartz (SiO₂), ettringite...." It seems there are extra letters.
Response5: Thank you for your careful reading. We have reviewed the sentence and corrected the typographical and formatting issues. The relevant statement has been revised for better clarity and accuracy.
Comments 6: And what about densities of the created samples with non-saying labeling (A, B, C, D)?
Response 6: Thank you for raising this insightful and critical point. We fully agree that the density of the samples is a key physical property for evaluating material performance. We acknowledge that the systematic measurement and reporting of density data for sample groups A, B, C, and D was overlooked during our experimental design, which is a clear limitation of this study.
To address this gap to the best of our current ability and to provide valuable reference information, we have included porosity data obtained through Mercury Intrusion Porosimetry (MIP) tests. Density and porosity are closely related (i.e., higher density typically corresponds to lower porosity). Our MIP data (see Section3.3.4. MIP Analysis in the manuscript) show a significant decreasing trend in total porosity with the incorporation of GO/NS, suggesting a potential increase in material density. This trend is consistent with the observed enhancement in mechanical properties, such as compressive strength.
While porosity data cannot fully substitute direct density measurements, we hope it offers useful insights for the current stage of this research. We have added this explanatory discussion to the relevant section of the manuscript. Furthermore, we commit to incorporating density measurement as a standard characterization method in our subsequent studies to ensure a more comprehensive and reliable dataset.
We sincerely appreciate your feedback, which has significantly improved the rigor and completeness of this work.
Reviewer 3 Report
Comments and Suggestions for Authors
Repeated reviewer's report on the paper titled “Performance Evolution and Multi-Scale Synergistic Mechanisms of Cement-Based Grouting Materials Modified with GO/NS Hybrids” for the MDPI Journal Materials
The paper “Performance Evolution and Multi-Scale Synergistic Mechanisms of Cement-Based Grouting Materials Modified with GO/NS Hybrids” was significantly improved after the first round of review. All the reviewers comments were addressed.
Author Response
We sincerely appreciate your meticulous review of this manuscript and the profound insights you have offered.